# Identification, Characterization, and Expression of a β-Galactosidase from Arion Species (Mollusca)

**DOI:** 10.3390/biom12111578

**Published:** 2022-10-27

**Authors:** Julia Thoma, David Stenitzer, Reingard Grabherr, Erika Staudacher

**Affiliations:** 1Department of Chemistry, University of Natural Resources and Life Sciences, Vienna, Muthgassse 18, 1190 Vienna, Austria; 2Department of Biotechnology, University of Natural Resources and Life Sciences, Vienna, Muthgasse 18, 1190 Vienna, Austria

**Keywords:** β-galactosidase, *Arion lusitanicus*, *Arion vulgaris*, mollusca, exoglycosidase, endoglycosidase, N-glycosylation

## Abstract

β-Galactosidases (β-Gal, EC 3.2.1.23) catalyze the cleavage of terminal non-reducing β-D-galactose residues or transglycosylation reactions yielding galacto-oligosaccharides. In this study, we present the isolation and characterization of a β-galactosidase from *Arion lusitanicus*, and based on this, the cloning and expression of a putative β-galactosidase from *Arion vulgaris* (A0A0B7AQJ9) in Sf9 cells. The entire gene codes for a protein consisting of 661 amino acids, comprising a putative signal peptide and an active domain. Specificity studies show exo- and endo-cleavage activity for galactose β1,4-linkages. Both enzymes, the recombinant from *A. vulgaris* and the native from *A. lusitanicus*, display similar biochemical parameters. Both β-galactosidases are most active in acidic environments ranging from pH 3.5 to 4.5, and do not depend on metal ions. The ideal reaction temperature is 50 °C. Long-term storage is possible up to +4 °C for the *A. vulgaris* enzyme, and up to +20 °C for the *A. lusitanicus* enzyme. This is the first report of the expression and characterization of a mollusk exoglycosidase.

## 1. Introduction

N-Glycosylation of proteins is one of the most complex forms of post-translational modification. Its accuracy is important for cellular processes, such as protein folding, secretion, cell adhesion, intracellular trafficking, recognition, and signaling [1,2]. The biosynthesis of N-glycans takes place in the ER and the Golgi, and is a species-specific tightly regulated interplay of several glycosyltransferases and glycosidases [3]. Most of the degradation is located in the lysosomes, and is mediated by highly specific endo- and exoglycosidases that act synergistically [4]. 

While these enzymes are well studied in mammals, plants, yeast, and bacteria, the knowledge about the glycosylation machinery of mollusks is still rudimentary, even when this phylum is evolutionary very successful. For more than 500 million years, mollusk species with very heterogenous morphology (gastropods, cephalopods, bivalvia) have populated freshwater, marine, and terrestrial habitats worldwide, executing important functions in the corresponding ecosystems in terms of waste disposal and cleaning. Others are used due to their nutritional value and shells [5]. Some of them are of medical relevance, as they act as intermediate hosts in parasite life cycles. 

Mollusks display a broad spectrum of glycosylation abilities shown in their heterogenous N- and O-glycan patterns. But so far, only a few of the involved glycosyltransferases and glycosidases have been described and characterized [6,7]. 

β-Galactosidases (β-Gal, EC 3.2.1.23) are a large family of exoglycosidases, which catalyze the cleavage of terminal non-reducing β-D-galactose residues (β1,3-, β1,4-, β1,6-) [8]. Furthermore, some of them are also able to catalyze transglycosylation reactions yielding galacto-oligosaccharides (GOSs) [9,10,11]. 

The best-studied β-galactosidase is certainly the one from *E. coli*, which is widely used in molecular biology as a reporter marker to screen gene expression through a technique called α-complementation. In *E. coli*, the protein is encoded by the *lacZ* gene, which is regulated by the lac operon in the presence/absence of lactose, and which requires Mg^2+^ and Na^+^ for maximal activity [12,13]. Further β-galactosidases, derived from other microorganisms (*Kluyveromyces*, *Penicillium*, *Lactobacillus*), have been investigated with regards to activity properties based on their structure [14,15,16,17]. Some of these enzymes are used on an industrial scale for the modification of oligosaccharides in food industry [18,19]. 

In multicellular organisms, β-galactosidases are located in the lysosomes or in the cytosol, depending on their function and substrate specificity, acting at acidic or neutral pH-optima. A change in the pH-environment has been found to play an important role in protein folding and, therefore, influences the activity properties [20,21].

Starting in the late seventies, snail extracts were found to be an interesting source of endo- and exoglycosidases. β-Galactosidases were detected and partly characterized from *Biomphalaria glabrata, Achatina achatina*, and *Pomacea canaliculata* [22,23,24,25]. However, in none of these studies were the enzymes expressed recombinantly. Similar to mammalian or plant enzymes, their function lies mainly in the degradation and modification of glycans. However, the snail glycosidases also seem to play a role in the snail immune response. In the plasma of *Biomphalaria glabrata*, the enzyme activity of several glycosidases, including β-galactosidase, was found to correlate with the progress of infection by *Schistosoma mansoni* [26]. 

Due to the enormous variety of glycosylation abilities of snails, and due to their continuous successful position in the ecosystem, snails are a valuable model for future investigations of glycosylation processes (for example—the interaction of parasites with their intermediate hosts). Detailed knowledge of the involved enzymes is therefore of essential importance. 

In this study we present, for the first time, the cloning and expression of a mollusk exoglycosidase (in particular, a β-galactosidase from *Arion vulgaris*). Furthermore, this enzyme was characterized and compared to a native β-galactosidase purified from *Arion lusitanicus*.

## 2. Materials and Methods

### 2.1. Materials 

*Arion lusitanicus* individuals were collected in September and October 2018 by the authors in their private gardens located in and around Vienna. Whole slugs were immediately frozen and stored at −80 °C until further use.

Q5/Taq DNA Polymerases, restriction enzymes, and T4 ligase were purchased from New England Biolabs (Frankfurt, Germany). All enzymes were used according to the supplier’s instructions. Primers and gBlock gene fragments were synthesized commercially by Sigma-Aldrich (Vienna, Austria) and Integrated DNA Technologies (Leuven, Belgium), respectively. A pACEBac1 vector was purchased from Geneva Biotech (Genève, Switzerland).

All other chemicals and molecular biology reagents were of the highest quality available. They were purchased from Sigma-Aldrich (Vienna, Austria), Merck (Darmstadt, Germany), Roth (Karlsruhe, Germany), Honeywell (Vienna, Austria), and ThermoFisher Scientific (Bonn, Germany), unless indicated otherwise.

Electrocompetent *E. coli* cells—NEB 5-alpha (NEB, Frankfurt, Germany)—were spread on Lysogeny Broth (LB) agar plates containing 15 µg/mL gentamycin and incubated overnight at 37 °C. Electrocompetent *E. coli* cells—DH10EMBacY cells (Geneva Biotech—Genève, Switzerland)—were cultivated on Lysogeny Broth (LB) agar plates containing 15 µg/mL gentamycin, 50 µg/mL kanamycin, 10 µg/mL tetracycline, 50 µg/mL IPTG, and 100 µg/mL X-Gal, and were incubated for 2 days at 37 °C. Electroporation was done using a MicroPulser from BIORAD.

*Spodoptera frugiperda* cells—Sf9 (ATCC, Manassas Virginia)—were grown in SFM4Insect media with L-Glutamine (HyClone Cytiva—Vienna, Austria) and kept at 27 °C [27]. Viable cell numbers were determined using the Vi-CellTM XR cell viability analyzer (Beckman Coulter—Vienna, Austria).

### 2.2. Galactosidase Purification from Arion Lusitanicus 

Approximately 3 g of *A. lusitanicus* viscera were homogenized in 150 mL 50 mM Tris/HCl buffer, pH 7.5, using an Ultra Turrax T25 (Janke & Kunkel IKA-Labortechnik—Staufen, Germany) at maximum speed. A batch ammonium sulphate precipitation was performed. The 30%-precipitate was discarded, and the 60%-precipitate was dissolved in 10 mL of 10 mM Na_2_HPO_4_ pH 7.0 + 1.2 M (NH_4_)_2_SO_4_, and loaded onto a hydrophobic interaction chromatography column (Octyl-sepharose CL-4B, 150 mL in the same buffer) [28]. A gradient from 1.2 M to 0 M (NH_4_)_2_SO_4_ was performed for elution. The obtained fractions with β-galactosidase activity were loaded onto a size exclusion column (Sephadex S200, 0.7 * 120 cm, in 50 mM Tris/HCl pH 7.5 + 0.02% NaN_3_) [28]. Fractions containing β-galactosidase were pooled, and the buffer was changed by ultrafiltration to 50 mM Tris/HCl pH 7.5, containing 0.1 M NaCl, 1 mM CaCl_2,_ and 1 mM MnCl_2_ for Concanavalin A (affinity chromatography (Concanavalin A coupled to Sepharose (Biorad, Vienna, Austria), 1 mL)). Elution was carried out with a stepwise increase of methyl-α-D-mannopyranoside from 15 mM to 100 mM, up to a final concentration of 500 mM. The fractions with β-galactosidase activity were pooled, and the buffer was changed to 50 mM NaCitrate (pH 4.6), containing 1 mM CaCl_2_ for β-galactosidase affinity chromatography ((p-aminobenzyl-1-thio-β-D-galactopyranoside; Sigma-Aldrich, Vienna, Austria), 1 mL, in 50 mM NaCitrate pH 4.6 containing 1 mM CaCl_2_). For elution, the following buffers were used sequentially: 50 mM NaCitrate (pH 4.6) containing 1 M NaCl, 50 mM Tris/HCl (pH 7.5), and, lastly, 50 mM Tris/HCl (pH 7.5) containing 1 M NaCl. The β-galactosidase activity containing fractions were used for mass spectrometry as well as biochemical characterization.

### 2.3. Mass Spectrometry of the Protein 

After the reduction with DTT (15 mM in 100 mM ammonium bicarbonate pH 7.0) and carbamidomethylation with iodoacetamide (30 mM) [29], the sample was digested with trypsin (Promega, sequencing grade), followed by a C18-SPE clean-up. The sample was loaded onto a LC (Dionex UltiMate 3000 Rapid, ThermoFisher Scientific - Bonn, Germany) equipped with nano-C18 column, coupled to an Orbitrap (ThermoFisher Scientific—Bonn, Germany, performed at IMBA, Vienna). A MS/MS ion search was performed using MASCOT. The files were searched against an in-house database (“snails”) and the reviewed uniprot database.

### 2.4. Expression of the Full-Length β-Galactosidase Gene from Arion Vulgaris 

Based on the mass spectrometry results of the native *A. lusitanicus* enzyme, the *A. vulgaris* β-galactosidase full-length gene (GenBank: A0A0B7AQJ9) was chosen and modified with a C-terminal hexahistidine-tag and a N-terminal gp64 secretion signal sequence (MVSAIVLYVLLAAAAHSAFA) [30]. 

The SacI and XbaI restriction sites were added to the sequence via primers (underlined in the primer sequences below). The recombinant gene was PCR amplified by using the forward primer 5′GATGATGAGCTCATGGTAAGTGCTATAGTGCTG3′ and the reverse primer 5′GATGATTCTAGATTAATGGTGGTGATGATGATG3′. The purified PCR fragment was ligated to the pACEBac1 vector in a 5:1 ratio. The correct insertion and gene sequence was verified by Sanger sequencing (Microsynth—Vienna, Austria). The recombinant plasmid (pACEBac1:β-gal) was amplified in NEB 5-alpha (NEB, Frankfurt, Germany), and 1 ng of the purified plasmid was used for integration into a MultiBac genome via Tn7 transposition in DH10EMBacY cells (Geneva Biotech, Genève, Switzerland). Midiprep of DH10EMBacY was performed by modifying the NucleoSpin Plasmid EasyPure kit protocol (Macherey-Nagel, Dueren, Germany) according to [31]. 5 µg of the recombinant β-gal construct were transfected to Sf9 insect cells (2 mL of 0.9 × 10^6^ cells/mL) using the FuGENE HD Transfection Reagent (Promega—Walldorf, Germany) in a 1:1.8 ratio, respectively. After 5 days, the seed stock was collected, and 100 μl was used for the production of the intermediate stock (8 × 10^6^ cells in 12 mL + 0.03% FBS). After 3–4 days, 200 μl of the intermediate stock was used for the production of the working stock (2 × 10^6^ cells/mL in 50 mL media + 0.04% FBS). For protein production, 100 μl of the working stock was added to 50 mL of 2 × 10^6^ cells/mL and incubated for 3 days. The recombinant β-gal protein was extracted from cell pellet fraction via the I-PER Cell Protein Extraction Reagent (ThermoFisher Scientific - Bonn, Germany), and further purified through immunoprecipitation using either protein G-plus or protein G-plus/Protein A-agarose beads (CALBIOCHEM, San Diego, CA, USA) linked to mouse anti Penta Histidine Tag:HRP monoclonal antibodies (BIORAD—Vienna, Austria) or magnetic beads (Abbkine—Wuhan, China), according to the supplier’s instructions. The purified recombinant protein was analyzed by SDS-Page and Western blot using the mouse anti Penta Histidine Tag:HRP monoclonal antibodies (1:2500, BIORAD—Vienna, Austria) following alkaline phosphatase conjugated anti-mouse IgG from goat (1:4000, Sigma—Aldrich, Vienna, Austria) [32].

### 2.5. Determination of Protein Content

Protein concentrations were determined by Micro-BCA protein assay (Pierce, Bonn, Germany) with bovine serum albumin as the standard.

### 2.6. Determination of β-Galactosidase Activity 

The analysis of the enzyme activity of β-galactosidase was based on a colorimetric assay using the artificial substrate 4-Nitrophenyl-β-D-galactopyranosid (pNP-β-Gal, Merck, Darmstadt, Germany). The reaction was performed in 50 µL containing 5 µL of enzyme solution, 20 µL of 0.9% (*w/v*) NaCl-solution, and 25 µL of pNP-β-Gal substrate (5 mM pNP-β-Gal in 0.1 M NaCitrat buffer, pH 4.5) at 37 °C for 2 h. The reaction was terminated by adding 200 µL of Glycine/NaOH (0.4 M, pH 10.4). The absorbance of the released p-nitrophenol was measured at 405 nm. For analysis of the biochemical parameters of the enzyme, the standard assay conditions using pNP-β-Gal as the substrate were modified as follows. For the determination of cation requirement, the standard assay was carried out without any cation addition, or in presence of 20 mM of EDTA, Mn^2+^, Mg^2+^, Ca^2+^, Co^2+^, Cu^2+^, Ni^2+^, or Ba^2+^. Chemical stability of the enzyme, optimal storage conditions, and pH-optimum were processed according to [28]. For storage stability in chemicals, the enzyme was incubated for approximately 16 h in 10% (*v/v*) or 20% (*v/v*) of methanol, acetonitrile, glycerol, or imidazole [50 mM or 100 mM]. For inhibition studies, the standard assay was performed in the presence of 6 mM or 12 mM monosaccharide (GlcNAc, GalNAc, Gal, Glc, Ara, Xyl, Fuc, Rha, Rib or Man). A time course was carried out by measuring the release of p-nitrophenol (up to 22 h in total). Kinetic data (K_M_-values) were acquired using pNP-β-Gal in a range from 0.2 mM to 12.5 mM in the standard incubation assay. 

Each assay was at least performed in duplicates with appropriate controls. All quantitative values were calculated using a calibration curve of p-nitrophenol. 

The substrate specificity was tested using different artificial pNP-sugars (pNP-α-Gal, pNP-α-Glc, pNP-β-Glc, pNP-α-GalNAc, pNP-β-GalNAc, pNP-β-GlcNAc, pNP-α-Fuc, pNP-α-Man, pNP-β-Man, and pNP-β-Xyl) under the same standard conditions as described above. To test activity towards natural substrates, di- and trisaccharides (lactose: Galβ1,4Glc; galacto-N-biose: Galβ1,3GalNAc; lacto-N-biose: Galβ1,3GlcNAc; 2-fucosyllactose: Fucα1,2Galβ1,4Glc) were labelled with 2-amino-benzoic acid [33], and were incubated at 37 °C over night with the enzyme preparation. Separation was carried out on reverse phase (ODS HYPERSIL, 4.6 × 250 mm, ThermoFisher Scientific—Bonn, Germany, solvent A: 0.2% (*v/v*) 1-butylamin, 0.5% (*v/v*) orthophosphoric acid, 1% (*v/v*) tetrahydrofuran in H_2_O, solvent B: solvent A/acetonitrile = 50/50 (*v/v*)), applying a linear gradient from 5–25% B in 17 min at a flow rate of 1.0 mL/min. Quantitative values were obtained by peak integration after fluorescence detection at ex/em 360 nm/425 nm. 

A diantennary N-glycan with terminal galactose residues (GalGal-OS: Galβ1,4GlcNAcβ1,2Manα1,3[Galβ1,4GlcNAcβ1,2Manα1,3]Manβ1,4GlcNAcβ1,4GlcNAc-oligosaccharide) was incubated under the same conditions as above with the enzyme preparations. The obtained molecular masses were determined by MALDI-TOF MS analysis on an Autoflex Speed MALDI-TOF (Bruker Daltonics, Germany) equipped with a 1000 Hz Smartbeam.II laser in positive mode using α-cyano-4-hydroxycinnamic acid as matrix. Spectra were processed with the manufacturer’s software (Bruker Flexanalysis 3.3.80).

The same diantennary N-glycan (GalGal-OS) was labelled with 2-amino-pyridine (GalGal-PA) [34], incubated as above, and separated on a Palpak column (4.6 × 250 mm, TaKaRa, San José, CA, USA), as previously described [35]. Quantitation was achieved using a fluorescence detector operating at ex/em 310 nm/380 nm.

A test for transglycosylation activity was carried out according to [36], with 5 mM pNP-β-Gal as donor and GlcNAc or Glc labelled with 2-amino-benzoic acid as the acceptors. The reaction was incubated at 37 °C over night with the enzyme preparation, and analysed as described above.

## 3. Results

### 3.1. Purification of β-Galactosidase from Arion Lusitanicus 

Viscera of *A. lusitanicus* were used for the purification of β-galactosidase. Performing several chromatographic steps, the enzyme was purified successfully [Table 1]. It was particularly difficult to remove other exoglycosidases (such as β-N-acetylglucosaminidase, α-mannosidase, β-mannosidase, and α-fucosidase) from the desired galactosidase. Finally, the total amount of enzyme was at about 0.10 U with a specific activity of 2.44 U/mg and a yield of 3.5%, which means a 62.83-fold purification.

The purified β-galactosidase fraction was used for biochemical analysis and, after tryptic digest, also for peptide analysis.

### 3.2. Recombinant β-Galactosidase from A. Vulgaris Expressed as Protein in Sf9 Insect Cells

Generally, there is a lack of genetic information regarding mollusks. Few of them are fully sequenced because, for most of them, just limited data are available. While for *A. vulgaris* at least some genetic data are available, there is no information regarding *A. lusitanicus*. Searching with the obtained peptides for homologies (Appendix A), one of the major hits was a predicted β-galactosidase from *Arion vulgaris*.

Furthermore, homology search was performed with active β-galactosidases from other organisms (*Homo sapiens, C. elegans, D. melanogaster*), which confirmed the identified candidate from *A. vulgaris* (A0A0B7AQJ9) as an interesting target [Figure 1].

The putative full-length β-galactosidase sequence from *A. vulgaris* (A0A0B7AQJ9) was selected and further modified to express and secrete the protein using the baculovirus expression system.

The gp64 leader sequence for secretion was added to the N-terminus, and a hexahistidine-tag for purification was added to the C-terminus of the sequence. The recombinant construct consisted of 688 amino acids with a molecular weight of approximately 77 kDa and a calculated isoelectric point (pI) of 5.8 [Figure 2].

The presence and activity of the expressed β-gal protein was analyzed in supernatant (secreted proteins) and lysate (soluble proteins). 

β-Galactosidase activity was present in the supernatant (secreted protein), as well as in lysate fractions (soluble, but not secreted protein) [Appendix A. Purification of the secreted protein (supernatant) over a HisTrap^TM^ excel column (1 mL, Cytiva—Vienna, Austria) was not successful, but by using immunoprecipitation of the lysate fraction, the pure active enzyme was obtained [Appendix A. 

### 3.3. Biochemical Parameters of Native and Recombinant β-Galactosidases from Two Arion Species

The optimal storage temperature for the recombinant β-galactosidase (*A. vulgaris*) was within a temperature range of −80 to + 4 °C, while the native enzyme from *A. lusitanicus* could be stored up to + 20 °C. The activity for both enzymes declined at storage temperatures above room temperature until the complete loss of activity at a temperature above 50 °C for *A. vulgaris* or 60 °C for *A. lusitanicus* [Appendix A]. However, 50 °C was the optimal reaction temperature for both enzymes in assays up to 2 h [Appendix A. The activity of the recombinant protein from *A. vulgaris* was not affected by lyophilization (data not shown). To investigate the storage stability in chemicals, the recombinant enzyme (*A. vulgaris*) was stored in methanol, acetonitrile, glycerol, or imidazole. The enzyme activity was highly affected by 20% (*v/v*) acetonitrile, as it drastically reduced activity to approximately 40% [Appendix A, as well as by imidazole [100 mM] and glycerol [10% (*v/v*)] during short term incubation (2 h, 37 °C). No influence on enzyme activity was observed by the addition of methanol [up to 40% (*v/v*)]. The native β-galactosidase from *A. lusitanicus*, however, was already influenced by the addition of 10% (*v/v*) acetonitrile, with an activity reduction to 40% during short term incubation. Altogether, only methanol was tolerated well by both species [Appendix A.

β-galactosidase activity was not dependent on divalent cations in both species. There was no loss of activity in the presence of EDTA. Addition of cations (Co^2+^, Mn^2+^, Mg^2+^, Ca^2+^, Ni^2+^, Ba^2+^) had no influence, except for Cu^2+^, which drastically reduced the activity to 17% (*A. lusitanicus*) and 68% (*A. vulgaris*) [Appendix A. 

The optimal pH for the native β-galactosidase from *A. lusitanicus*, as well as for the recombinant *A. vulgaris* enzyme, was in the range of pH 3.5–4.5 using acetat or citrat as buffer salts [Figure 3, Appendix A.

In consideration of inhibiting substances, the activity of the recombinant protein (*A. vulgaris*) was tested in the presence of monosaccharides (GlcNAc, GalNAc, Gal, Glc, Ara, Xyl, Fuc, Rha, Rib, and Man). Thereby, product inhibition by galactose was detected by the addition of 6 mM and 12 mM galactose, which resulted in an approximately 25% and 40% reduction of activity, respectively. All other monosaccharides did not show any inhibitory effects [Appendix A.

Analysis of enzyme kinetics using the pNP-β-Gal substrate at a pH of 4.5 revealed a K_M_ of 8.3 mM and a v_max_ of 0.002 μmol/min for *A. lusitanicus*, and a K_M_ of 3.2 mM and a v_max_ of 0.0002 μmol/min for *A. vulgaris*.

### 3.4. Determination of Substrate Specificity of β-Galactosidase 

Other artificial chromogenic pNP-substrates, such as pNP-α-Gal, pNP-α-Glc, pNP-β-Glc, pNP-α-GalNAc, pNP-β-GalNAc, pNP-β-GlcNAc, pNP-α-Fuc, pNP-α-Man, pNP-β-Man, and pNP-β-Xyl were tested in order to investigate the substrate specificity of the β-galactosidase from *A. vulgaris*. Using this selection, the enzyme cleaved pNP-β-Gal and, to a minor amount (37%), pNP-α-Glc. To specify the β-galactosidase specificity regarding the galactose position (β1,3- or β1,4-) and the identity of the other sugar involved in the linkage, native substrates were evaluated. The diantennary oligosaccharide (GalGal-OS, Galβ1,4GlcNAcβ1,2Manα1,3[Galβ1,4GlcNAcβ1,2Manα1,3]Manβ1,4GlcNAcβ1,4GlcNAc-oligosaccharide), or its pyridylaminated variant (GalGal-PA) with the terminal galactoses bound (β1,4 to N-acetylglucosamine residues), were appropriate substrates (Figure 4).

2-AB labelled lactose (Galβ1,4Glc) was cleaved, while the two disaccharides containing a β1,3-linkage (2-AB-lacto-N-biose: Galβ1,3GlcNAc and 2-AB-galacto-N-biose: Galβ1,3GalNAc) were not cleaved at all. That suggested that the enzyme might be rather specific for β1,4-linkages. Surprisingly, we also detected a clear cleavage of 2-AB-labelled 2-fucosyllactose (Fucα1,2Galβ1,4Glc) substrate, also indicating an endo-glycosidase activity.

Furthermore, because some β-galactosidases also catalyze a transglycosylation reaction, we tested activity with pNP-β-Gal as the donor, and GlcNAc or Glc as acceptors, according to [36]. However, no transglycosylation activity was detectable under the chosen conditions.

## 4. Discussion and Conclusions

β-Galactosidases (EC 3.2.1.23) are a large family of exoglycosidases which catalyze single cleavage reactions that remove terminal galactose residues from N- and O-glycans, hydrolyze β-galactosides into monosaccharides, or produce galacto-oligosaccharides (GOSs) by a transglycosylation reaction [10,36]. They can be found throughout the animal kingdom, as well as in plants, yeast, and bacteria [18,37]. In mollusks, several studies were conducted to analyze and characterize β-galactosidases of different species [23,38,39,40]. However, compared to the β-galactosidases from *E. coli* or humans, only little is known about these enzymes in mollusks. 

In search for the β-galactosidase from Arion, we started by purifying the enzyme form *A. lusitanicus* through several chromatographic steps. The sample was further analyzed with a LC coupled to an Orbitrap (Thermo), and blasted against the in-house “snail” databank. Based on a homology search with the fragments obtained from the purified enzyme, we successfully expressed the predicted β-galactosidase enzyme from *A. vulgaris* (A0A0B7AQJ9, MASCOT score: 2221.1) in Sf9 insect cells. 

The activity and specificity of the full-length recombinant *A. vulgaris* enzyme was compared with the purified native *A. lusitanicus* enzyme using the artificial pNP-β-Gal and several native substrates.

In terms of pH-optimum, the β-galactosidase enzymes from *A. lusitanicus* and *A. vulgaris* were mostly active in acidic environments ranging from pH 3.5 to 4.5, which correlates with other β-galactosidases from mollusks, which range from pH 3.2 to 5.6 [23,38,39,40]. In general, β-galactosidases are known to work in a relatively broad pH range. The enzymes from fungi, on the one hand, prefer more acidic environments with a pH-value starting at 2.5 and extending to 7.0 [41]. On the other hand, bacterial β-galactosidases favor nearly neutral environments, and hence act best between pH 7.0 and 7.5 [17]. Moreover, in mammals, the specific pH of the protein depends on the enzyme’s localization, e.g., lysosomal or cytosolic, and varies between pH 3.0 and 6.0 [42]. Hence, we concluded that in Arion the acid optimum of our protein might indicate its localization in lysosomes rather than in the cytoplasm. 

The influence of cations on β-galactosidases from different species is very heterogenous. In the microorganism *Bacillus stearothermophilus*, most cations had no effect on the enzyme’s activity. Heavy metals—Cu^2+^, as well as Fe^2+^, Zn^2+^, Cu^2+^, Pb^2+^, and Sn^2+^—were shown to inhibit the enzyme drastically. In contrast, the enzyme from *E. coli* showed the need for Mg^2+^ and Mn^2+^ for its activity [43,44]. In addition, kinetic studies on the β-galactosidase of *E. coli* showed that Mg^2+^ is more important for the catalytic process, rather than for substrate binding or the formation of the quaternary structure [45]. Regarding mammalian β-galactosidase, the necessity of MgCl_2_ on the enzyme in rat liver was shown [46]. Nevertheless, the influence of ions on mammalian β-galactosidase, and especially those from mollusks, has received only little attention compared to those from microorganisms.

In regards to cation additives, we identified an inhibitory effect on our snail enzymes by Cu^2+^, which might be attributed to the metal-catalyzed oxidation of some critical amino acid residues, e.g., histidine or cysteine [47]. Other ions, such as Mn^2+^, Ba^2+^, Mg^2+^, Ca^2+^, Co^2+^, and Ni^2+^, had no significant effect on the enzymes’ activities from both Arion species. Hence, the β-galactosidase from *A. lusitanicus* and *A. vulgaris* were independent in terms of metal ions, supporting previous data from *Achatina balteata* [40]. 

β-Galactosidases from various sources exhibit significant differences in their temperature optimum. The fungal enzymes work in ranges from 4 to 67 °C, while most bacterial β-galactosidases have optima between 40 and 65 °C [18,19,48]. For both of our snail enzymes, we identified an optimal reaction temperature of 50 °C, which also correlates with other β-galactosidases from mollusks [23]. Moreover, the recombinant protein from *A. vulgaris* was active for up to 22 h at 37 °C. The addition of several chemicals lowered the activity drastically. Especially the negative effect of imidazole has to be emphasized, as it plays an essential role in the purification of His-tagged proteins. From all chemicals tested, only methanol did not influence the enzyme’s activity.

Analysis of enzyme activity revealed a K_M_ of 8.3 mM for *A. lusitanicus* and a K_M_ of 3.2 mM for *A. vulgaris* using the pNP-β-Gal substrate. Both K_M_ values are similar compared to other β-galactosidases using the same artificial substrate as shown for *E. coli* with a K_M_ of 0.093 mM [49], the bovine β-galactosidase with a K_M_ of 2.5 mM, or the fungi β-galactosidase with a K_M_ of 1.0 mM [17].

Both enzymes (native and recombinant) highly favored the artificial pNP-β-Gal, but also showed low activity on pNP-α-Glc. Activity was further tested on native β1,3- and β1,4- substrates. Exclusively β1,4- linkages were cleaved. Terminal galactose residues on biantennary glycans were clearly preferred over lactose by the exoglycosidase activity. We further detected some endoglycosidase activity cleaving 2-fucosyllactose. No transglycosylation activity could be detected.

In conclusion, for the first time an exoglycosidase from mollusk origin was recombinantly expressed, characterized, and compared to a purified native enzyme originating from the same genus (Arion). The snail enzyme was found to be specific towards β1,4-linkages, and displays exo- as well as endoglycosidase activity. Our results showed that both enzymes, the recombinant from *A. vulgaris* and the native from *A. lusitanicus*, are akin in their biochemical parameters.

## Figures and Tables

**Figure 1 biomolecules-12-01578-f001:**
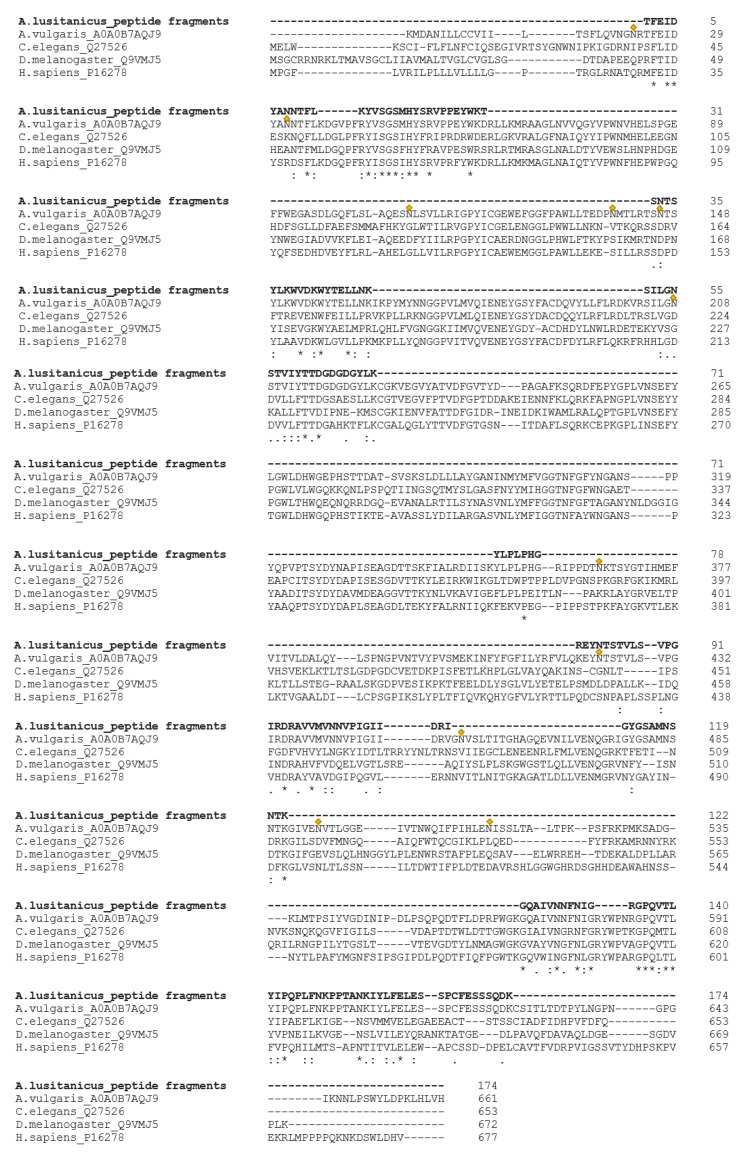
Peptides found in *A. lusitanicus* aligned to active β-galactosidase from *Homo sapiens* (P16278), *D. melanogaster* (Q9VMJ5), *C. elegans* (Q27526), and putative β-galactosidase from *A. vulgaris* (A0A0B7AQJ9). Yellow rhomb indicates putative N-glycosylation sites in *Arion vulgaris* according to the N-X-S/T consensus sequence. (*) marks identical amino acid residues, (:) related amino acid residues, and (.) predominantly the same amino acid residues.

**Figure 2 biomolecules-12-01578-f002:**
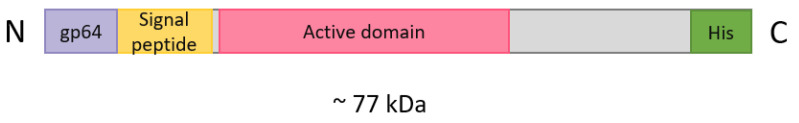
Schematic structure of the recombinant full-length β-gal from A. vulgaris (GenBank: A0A0B7AQJ9). The full-length protein consists of a putative N-terminal signal peptide (yellow), a putative active domain (red), and further gene sequence (grey). A gp64 leader sequence (purple) for secretion out of Sf9 insect cells was added to the N-terminus, and a 6x His-tag (green) for purification was added to the C-terminus of the protein. A short 3 amino-acid linker was added to the gene at the N- and C-terminus to connect gp64 leader sequence and 6x His-tag (not shown in the figure).

**Figure 3 biomolecules-12-01578-f003:**
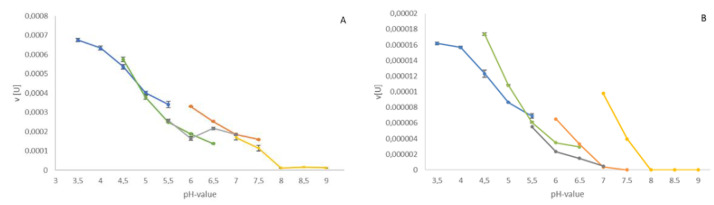
pH pattern of Arion using the following buffer systems: acetate/NaOH (blue), citrate/NaOH (green), MES (grey), phosphate (orange), and Tris/HCl (yellow). (**A**) *A. lusitanicus*; (**B**) *A. vulgaris*. Data points represent mean values of duplicate measurements in units with corresponding standard deviation. Negative values, due to conversion into units, were set to zero. For experimental details see Section 2.6.

**Figure 4 biomolecules-12-01578-f004:**
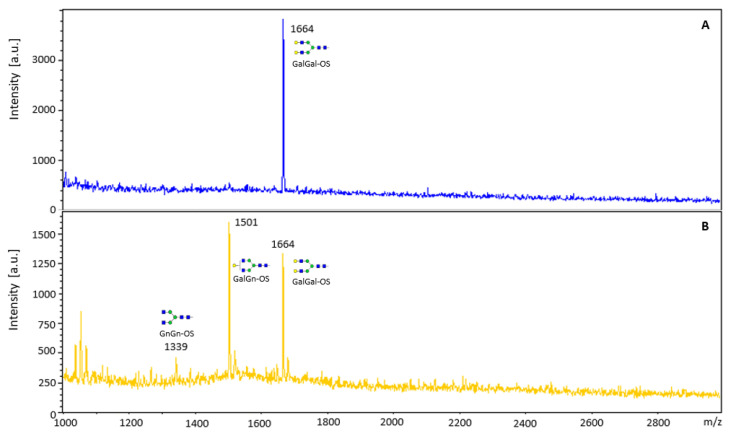
MALDI-TOF analysis of the release of galactose from GalGal-OS using the recombinant β-galactosidase from *A. vulgaris.* (**A**) GalGal-OS substrate; (**B**) GalGal-OS incubated with β-galactosidase.

**Table 1 biomolecules-12-01578-t001:** Purification protocol of β-galactosidase from *A. lusitanicus*.

	Total Protein [mg]	Total Enzyme [U]	Specific Activity [U/mg]	Purification Fold	Yield [%]
After precipitation	72.74	2.82	0.04	1.00	100.00
After HIC	2.89	1.62	0.56	14.40	57.31
After S200	0.63	0.53	0.84	21.76	18.83
After Con A	0.22	0.21	0.99	25.64	7.62
After Gal-affinity chromatography	0.04	0.10	2.44	62.83	3.51

## Data Availability

Not applicable.

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
