# Peer review of "Identification, Characterization, and Expression of a β-Galactosidase from Arion Species (Mollusca)"

_biomolecules, 2022, doi:10.3390/biom12111578_

Round 1

Reviewer 1 Report

Thoma and co-workers performed the identification and initial biochemical characterization – including cloning, recombinant production in Sf9 cells, and purification – of a β-galactosidase from A. vulgarisThe activity was shown with pNP surrogates and a selection of oligosaccharides. Interestingly, the β-galactosidase also showed endo-glycosidase activity. The authors employ a broad repertoire of established methods and use them in a comprehensible manner. The manuscript is written concisely. However, the authors are asked to implement the following to improve the manuscript:

(1) Include data from replicates of pNP-based colorimetric assays – either in the manuscript or the supplementary information (SI).

(2) Add minimal statistical analysis to Figure 3. - Do the data points represent mean values of replicate measurements? Please, add, for example, the standard deviation in the graph and minimal experimental details to the figure legend.

(3) Please, add additional references – for example, the carbamidomethylation with iodoacetamide in the material/methods section – to address a broader audience of readers.

Furthermore, could the authors answer the following minor questions:

(1a) The Km values for pNP-β-Gal were 8.3 mM and 4.5 mM for A. lusitanicus and A. vulgaris, respectively. Have the authors determined Km values for non-surrogate compounds as well? Can such Km values be related to physiological concentrations or relevance for Arion species (or related mollusks)?

(2a) Can the quality of pictures in the SI be improved?

(3a) Line 53–54: Here, β-galactosidases from different species (Kluyveromyces, Penicillium, E. coli) are mentioned. However, the one from E. coli has already been introduced. Maybe remove E. coli and replace by yeast, for example.

(4a) Why was the endo-glycosidase activity not highlighted further?

Author Response

The authors thank the reviewers for their helpful comments and suggestions. Please find enclosed in detail our changes regarding each comment.

Reviewer 1

  • Include data from replicates of pNP-based colorimetric assays – either in the manuscript or the supplementary information (SI).

Additional images regarding the biochemical parameters are given in the SI. As an example, we added an additional table with the raw data for the pH-optimum. Please note that these data are the absorbances which were converted into units for the graphs, according to the other reviewer´s suggestion.

  • Add minimal statistical analysis to Figure 3. - Do the data points represent mean values of replicate measurements? Please, add, for example, the standard deviation in the graph and minimal experimental details to the figure legend.

Explanation is added in the legends. Standard deviation is given in the graph.

  • Please, add additional references – for example, the carbamidomethylation with iodoacetamide in the material/methods section – to address a broader audience of readers.

Additional reference Geisslitz et al. is added in section 2.3. Furthermore, more citation is given within section 2.2.

Furthermore, could the authors answer the following minor questions:

(1a) The Km values for pNP-β-Gal were 8.3 mM and 4.5 mM for A. lusitanicus and A. vulgaris, respectively. Have the authors determined Km values for non-surrogate compounds as well? Can such Km values be related to physiological concentrations or relevance for Arion species (or related mollusks)?

No, we did not determine the Km for non-surrogate substrates. Currently we are working on a reliable and exact quantification for those substrates. Most other current visualization systems are suboptimal: unlabeled di- or trisaccharides (e.g. quantification via thin layer chromatography) lack sensitivity and accuracy, fluorescent/UV-detectable labels may affect the response in ways not yet studied.

As far as we know, there are no physiological studies on Arion species so far.

(2a) Can the quality of pictures in the SI be improved?

In the meantime, several journals reject a paper when they see any modification in appearance of figures, any use of “photo-shop”. Therefore, we avoided any image processing. 

(3a) Line 53–54: Here, β-galactosidases from different species (Kluyveromyces, Penicillium, E. coli) are mentioned. However, the one from E. coli has already been introduced. Maybe remove E. coli and replace by yeast, for example.

  1. coli was replaced by Lactobacillus.

(4a) Why was the endo-glycosidase activity not highlighted further?

In this paper we focused on the purification and expression cloning of the enzyme and its basic characterization, especially comparing the native and the recombinant enzyme. The endo-glycosidase activity is an interesting issue which needs a large number of potential substrates, which would go beyond the scope of this paper.  

Author Response

The authors thank the reviewers for their helpful comments and suggestions. Please find enclosed in detail our changes regarding each comment.

Reviewer 2

  1. According to journal’s instructions for authors, some of the references in the text are
    not properly written. For example: Line 44: [6, 7] should be written without spaces between the numbers [6,7]

Corrected.

  1. In Materials and Methods section (2.1. Materials) the date on which the samples were
    collected should be provided. Additionally, the location where the samples were
    collected should be explained in more detail. Were the slugs collected in a park
    outside Vienna or in the city itself? Finally, details of how the slugs and the viscera
    samples were stored prior to use should be provided.

Details of sample collection and storage are given. 

  1. In Results section (3.3. Biochemical parameters of native and recombinant
    b-galactosidases from two Arion species) data regarding chemical stability of the
    enzyme, optimal storage and activity conditions at different temperatures and
    enzymatical activity with different cations should be plotted. Additionally, these data
    should be explained in more detail. For example, when authors state that a severe
    activity loss is observed, more detailed data must be provided. What percentage of
    activity is retained with respect to the maximum activity observed under optimal
    conditions? This information is in fact provided when talking about product inhibition
    by galactose.

We added the graphs of optimal reaction temperature, optimal storage temperature, influence of chemicals, influence of divalent cations and influence of the addition of monosaccharides to the Supplements.

  1. In Results section (3.3. Biochemical parameters of native and recombinant b-galactosidases from two Arion species) authors claim that optimal incubation temperature for short assays was 50 ºC. This should be referred as optimal reaction temperature in order to avoid confusion with the storage stability mentioned in this section.

Wording was changed and furthermore it is explained in the legends of the new Figs S3 and S4.

  1. In Results section (3.3. Biochemical parameters of native and recombinant b-galactosidases from two Arion species) data in Figure 3 should be displayed in a different manner. On the y-axis instead of absorbance units, enzyme activity data should be given. Preferably, the activity data should be relative to the maximum activity detected in these experiments. The figure caption should be also revised and corrected since the symbols cannot be seen.

The graph was changed according to the reviewer´s suggestions. The y-axis is now changed to units and the symbols are replaced by coloured lines.

  1. Similarly, in 3.4. Determination of substrate specificity of b-galactosidase section, data regarding the substrate specificity should be plotted and activity values (at least for best substrates) must be provided. In this sense, Figure 4 (MALDI-TOF analysis) should be provided as supplementary data.

Substrate specificity studies were carried out with a number of different methods Activity measurement using artificial pNP-labelled substrates is easy and a reliable quantification is possible. Most other current visualization systems are suboptimal: unlabeled di- or trisaccharides (e.g. quantification via thin layer chromatography) lack sensitivity and accuracy, fluorescent/UV-detectable labels may affect the response in ways not yet studied. Currently we are working on a reliable and exact quantification for those substrates.

For now, we do not want to quantify, but just give a Yes/No information.

MALDI-TOF is the best proof for the activity of the enzyme, therefore we kept the related Figure in the main part.

  1. All in all, the biochemical characterization section is quite confusing, since it is not clear which data refer to A. lusitanicus and which to A. vulgaris. These whole section should be rewritten in a clearer manner.

Rephrasing was done.

  1. Finally, English language and style must be checked. Some spelling errors have been
    found in the manuscript (e.g., line 285).

Paper was proof read. We hope, we found all errors.

Round 2

Reviewer 2 Report

I have gone through the revised manuscript re-submitted by Thoma et al., and several improvements have been done in the text. Reviewer suggestion have been taken into consideration and, thus, results are presented in a much clearer manner in this new version. However, I still recommend minor revision prior to publication since some spelling errors have been found in the manuscript (e.g., line 286, 295 and 296). In this sense, English language and style must be carefully checked.